# Comparison of different Lunit INSIGHT CXR software versions when reading chest radiographs for tuberculosis

**Andrew J. Codlin** [1,2*], **Luan N. Q. Vo**[1,2], **Thang P. Dao**[1], **Rachel J. Forse**[1,2], **Ha T. M. Dang**[3], **Lan H. Nguyen**[3], **Hoa B. Nguyen**[4], **Luong V. Dinh**[4], **Kristi Sidney Annerstedt**[2], **Johan Lundin** [2,5], **Knut Lönnroth**[2]

**1** Friends for International TB Relief, Ha Noi, Viet Nam, **2** Department of Global Public Health, Karolinska Institutet, Stockholm, Sweden, **3** Pham Ngoc Thach Hospital, Ho Chi Minh City, Viet Nam, **4** National Lung Hospital, Ha Noi, Viet Nam, **5** Institute for Molecular Medicine Finland (FIMM), Helsinki Institute of Life Science (HiLIFE), University of Helsinki, Helsinki, Finland

* andrew.codlin@tbhelp.org; andrew.james.codlin@ki.se

## Abstract

New versions of computer-aided detection (CAD) software for chest X-ray (CXR) interpretation during tuberculosis (TB) screening are regularly released which purport to have incremental performance gains. No studies have independently assessed differences in software performance between the World Health Organization recommended INSIGHT CXR software (Lunit, South Korea). A well-characterized Digital Imaging and Communications in Medicine (DICOM) test library was compiled using data from a community-based TB screening initiative in Ho Chi Minh City, Viet Nam. The performance of Lunit CAD software versions 3.1.0.0 and 3.9.0.1 (newer version) were compared by measuring the area under the receiver operating characteristic curve (AUC), stratified by key clinical and demographic variables and using Xpert MTB/RIF Ultra (Ultra) test results as the reference standard. Median abnormality scores were compared using the Wilcoxon signed-rank test and performance characteristics were compared at clinically-relevant cut-off thresholds (e.g., 90% sensitivity) between the versions. The DICOM test library contained 2,708 participants, of whom 10.3% had a *Mycobacterium tuberculosis* (MTB) positive Ultra test result. The newer software version had a significantly higher AUC than its predecessor (AUC 0.76 vs 0.78, p = 0.029), and performed significantly better among people with a past history of TB (AUC 0.67 vs 0.73, p = 0.003), older individuals (0.75 vs 0.77, p = 0.040) and males (0.73 vs 0.76, p = 0.008). When using an cut-off threshold optimized for the older software version, the newer software was significantly less accurate than its predecessors. However, when the cut-off threshold was re-calibrated, there were no significant differences in sensitivity and specificity between the software versions. Although INSIGHT CXR v3.9.0.1 has some significantly improved performance characteristics compared to its predecessor, further studies should assess how these performance differences translate into real-world improvements during TB screening. As new CAD software versions are rolled out, cut-off thresholds must be re-calibrated to ensure the continued accuracy of CXR interpretation.

**Data availability statement:** This evaluation's data set and DICOM test library are publicly available in a Dryad repository (https://doi.org/10.5061/dryad.5tb2rbpcv).

**Funding:** Funding for the programmatic active case finding which was the source of the data for this evaluation's DICOM test library was funded by the European Commission's Horizon 2020 programme (agreement number: 733174), the Stop TB Partnership's TB REACH initiative (agreement numbers: STBP/TBREACH/GSA/W6-42 & STBP/TBREACH/GSA/W6SU-09) and USAID (agreement number: 72044020FA00001). These funders had no role in study design, data collection and analysis, decision to publish, or preparation of the manuscript.

**Competing interests:** The authors have declared that no competing interests exist.

## Author summary

Computer-aided detection (CAD) software are a new class of technology which can read and interpret medical images in the absence of a radiologist. In this manuscript, we describe the performance gains between two successive versions of a CAD software which can be used to identify people in need of further testing for tuberculosis (TB) during a chest X-ray screen. The new software version is more accurate than its predecessor in several key metrics, but it is unclear whether this performance gain will translate into clinically relevant improvements during prospective TB screening. Transitioning from one software version to another will not be as straightforward as initially thought. The new software version cannot use the same deployment settings as its predecessor version and will need to be re-calibrated in order to maintain the accuracy of CXR interpretation.

## Introduction

Chest X-ray (CXR) screening is a highly sensitive approach for detecting tuberculosis (TB). Modern TB prevalence surveys from Africa [1] and Asia [2] have consistently shown that a large proportion (40–79%) of people with infectious TB do not report classical TB symptoms (e.g., prolonged cough), and can only be indicated for diagnostic testing via the use of this more sensitive screening test. As a result, there has been a recent increase in interest and use of CXR screening as a TB case finding strategy, particularly in high TB burden settings [3].

New digital radiography systems, including ultraportable devices [4], have lowered past access barriers, including the stationary nature of fixed X-ray systems, upfront and recurring costs, and concerns about excess radiation, so that CXR screening for TB can be implemented in affected communities and remote settings. However, a key remaining barrier to the widespread adoption of CXR screening for TB, is the global dearth of quality-assured radiologists, particularly in low- and middle-income countries where the TB epidemic is concentrated [5,6]. In light of this human resource gap, the World Health Organization's (WHO's) latest TB screening guidelines recommend the use of three computer-aided detection (CAD) software (CAD4TB [Delft Imaging, The Netherlands]; INSIGHT CXR [Lunit, South Korea]; and qXR [Qure.ai, India]) in place of human readers for the interpretation of CXR images during TB screening [7]. These CAD solutions are having a real-world impact on TB screening programs, resulting in increased TB detection, particularly asymptomatic TB, and reduced consumption of diagnostic tests and radiologist workloads [8–11]. Since the WHO guidelines were released, multiple other CAD software platforms have been developed which match the performance of the three WHO-recommended platforms in independent evaluations [12–18].

CAD software are regularly trained with additional clinical data and their underlying algorithms are updated with each newly released software version, while maintaining the same branding. This contrasts with TB diagnostic assays, whose developers infrequently release new assay versions and usually brand the updated version differently to indicate superior performance (e.g., Xpert MTB/RIF [Xpert] to Xpert MTB/RIF Ultra [Ultra], Truenat MTB [TruMTB] to Truenat MTB Plus [TruMTB-Plus], or QuantiFERON-TB Gold In-Tube [QFT-GIT] to QuantiFERON-TB Gold Plus [QFT-Plus]). To date, only three independent studies have compared the performance of successive versions of WHO-recommended CAD software [19–21]. They found that the newest versions of CAD4TB and qXR do indeed have significantly better performance characteristics than their predecessors. However, these studies have highlighted differences in abnormality score distributions between software versions and

the need for re-assessment of cut-off thresholds to ensure that CAD software performance is maintained. No published studies have documented the performance of successive versions of the third WHO-recommended CAD software: INSIGHT CXR.

In this evaluation, we compare the performance of INSIGHT CXR v3.1.0.0 and the newer v3.9.1.0 using a Digital Imaging and Communications in Medicine (DICOM) test library compiled using data from a programmatic active case finding (ACF) initiative in Viet Nam.

## Results

This evaluation's DICOM test library contains a total of 2,708 individuals, of whom 280 (10.3%) had a *Mycobacterium tuberculosis* (MTB) positive result on the Ultra assay (Table 1). The test library contains more males (n = 1,814) than females (n = 894), and males have a significantly higher Ultra test positivity rate (n = 228, 12.6% vs n = 52, 5.8%; p < 0.001). 79.9% of the test library is comprised of older individuals (≥55 years), yet the library's younger individuals have a significantly higher Ultra test positivity rate (15.8% vs 9.0%, p < 0.001). 44.0% of the test library has a positive WHO four-symptom screen (W4SS, comprising any one symptom of cough, fever, weight loss and/or night sweats of any duration), and these symptomatic individuals have a significantly higher Ultra test positivity rate (12.3% vs 8.8%, p = 0.002). Despite this, 47.5% of the total number of individuals with a MTB positive test result are asymptomatic (W4SS negative). 7.2% of the test library reported having contact with someone who had TB in their household or in the community, and these individuals had a significantly higher Ultra test positivity rate than non-contacts (15.5% vs 9.9%, p = 0.015). 35.2% of the test library was treated for TB in the past, 9.8% reported having diabetes and 0.2% reported having an HIV infection; however, these risk factors were not associated with significantly increased Ultra test positivity rates in this test library. There are no significant differences in Ultra test positivity rates across the radiography systems.

Fig 1 shows the receiver operating characteristic (ROC) curves for the two Lunit CAD software versions. The v3.1.0.0 software achieved an area under the curve (AUC) of 0.76, whereas the newer v3.9.0.1 software achieved a significantly higher AUC of 0.78 (p = 0.029). AUC performance gains appear to be concentrated between the 25%–75% sensitivity range of the ROC curve; in the >75% sensitivity range, the curves for both software are mostly overlapping. There was no significant difference in the AUCs between the CAD software versions in individuals without a past history of TB (0.81 vs 0.81, p = 0.644). However, the AUC for the newer v3.9.0.1 software was significantly higher among individuals with a past history of TB (0.67 vs 0.73, p = 0.003), with gains again concentrated in the 25%–75% sensitivity range of the ROC curve. Similarly, there were no significant differences AUCs between the CAD software versions in younger individuals (0.78 vs 0.80, p = 0.425) and females (0.80 vs 0.79, p = 0.875), but the newer v3.9.0.1 software's AUCs were significantly higher among older individuals (0.75 vs 0.77, p = 0.040) and males (0.73 vs 0.76, p = 0.008).

Table 2 shows a comparison of AUCs between the Lunit CAD software versions for sub-cohorts with combinations of demographic and clinical factors. The newer v3.9.0.1 software achieved a significantly higher AUC among older males with a past history of TB that its predecessor (0.63 vs 0.72, p < 0.001). In the remaining seven sub-cohorts, no signfiicant differences in AUCs between the software versions were recorded.

The median Lunit CAD abnormality score was significantly higher with the newer v3.9.0.1 software, for all individuals regardless of their Ultra test result (Table 3). 26.4% of people with a MTB positive Ultra test result had a Lunit CAD abnormality score <0.50 (default cut-off threshold) using the v3.2.0.0 software, compared to 10.4% with the newer v3.9.0.1 software, indicating that there is still a sizable proportion of individuals with MTB positive results who have low abnormality scores with the newer software version.

**Table 1. Test library demographic and clinical details.**

| | All Individuals | Xpert MTB/RIF Ultra test result | | p-value |
|---|---|---|---|---|
| | | MTB negative | MTB positive | |
| All individuals | 2,708 | 2,428 (89.7%) | 280 (10.3%) | N/A |
| **Demographic factors** | | | | |
| Gender | | | | |
| Male | 1,814 (67.0%) | 1,586 (87.4%) | 228 (12.6%) | **<0.001** |
| Female | 894 (33.0%) | 842 (94.2%) | 52 (5.8%) | |
| Age | | | | |
| 18–54 years | 544 (20.1%) | 459 (84.4%) | 85 (15.6%) | **<0.001** |
| ≥55 years | 2,164 (79.9%) | 1,969 (91.0%) | 195 (9.0%) | |
| **Presence of TB symptoms** | | | | |
| Cough | 1,090 (40.3%) | 956 (87.7%) | 134 (12.3%) | **0.006** |
| Fever | 94 (3.5%) | 77 (81.9%) | 17 (18.1%) | **0.012** |
| Weight loss | 193 (7.1%) | 158 (81.9%) | 35 (18.1%) | **<0.001** |
| Night sweats | 74 (2.7%) | 66 (89.2%) | 8 (10.8%) | 0.893 |
| W4SS* positive | 1,192 (44.0%) | 1,045 (87.7%) | 147 (12.3%) | **0.003** |
| **TB risk factors** | | | | |
| Contact of person with TB | 194 (7.2%) | 164 (84.5%) | 30 (15.5%) | **0.015** |
| Past history of TB | 952 (35.2%) | 850 (89.3%) | 102 (10.7%) | 0.637 |
| Diabetes | 266 (9.8%) | 232 (87.2%) | 34 (12.8%) | 0.168 |
| HIV | 5 (0.2%) | 4 (80.0%) | 1 (20.0%) | 0.478 |
| **Radiography system** | | | | |
| JPI Healthcare | 408 (15.1%) | 364 (89.2%) | 44 (12.1%) | 0.132 |
| DRTECH | 1,332 (49.2%) | 1,181 (88.7%) | 151 (12.8%) | |
| Konica Minolta | 968 (35.7%) | 883 (91.2%) | 85 (9.6%) | |

*W4SS = World Health Organization four-symptom screen.

Table 4 shows the performance of Lunit CAD software versions at different abnormality score cut-off thresholds, above which CXR images were classified as abnormal. At a cut-off threshold of 0.217, the v3.1.0.0 software achieves a 90.0% sensitivity and a 45.8% specificity. Using this same cut-off threshold, the newer v3.9.0.1 software achieves a significantly higher sensitivity of 96.4%, but a significantly lower specificity of 23.1% (p < 0.001). This would translate to a +36.4% increase in diagnostic testing, for a corresponding +6.7% increase in yield of TB. When a second cut-off threshold is selected for the newer v3.9.0.1 software so that it achieves a 90% sensitivity (0.480), there is no statistical difference with the newer software's specificity (43.2%) or the additional Ultra tests indicated (+4.1%) compared with the v3.1.0.0 software version (p = 0.148) at its original threshold.

## Discussion

This is the first independent evaluation of Lunit INSIGHT CXR software versions. Our results show that the newer software version has significantly improved performance to detect TB-related abnormalities on CXR images compared to its predecessor, with the gains being concentrated among older males with a history of TB. This finding is encouraging, as past comparative CAD software evaluations have shown that CXR interpretation in people with a past history of TB is consistently less accurate for almost all CAD software platforms [12,14]. However, the real-world implications of this performance gain must be further investigated. Between successive

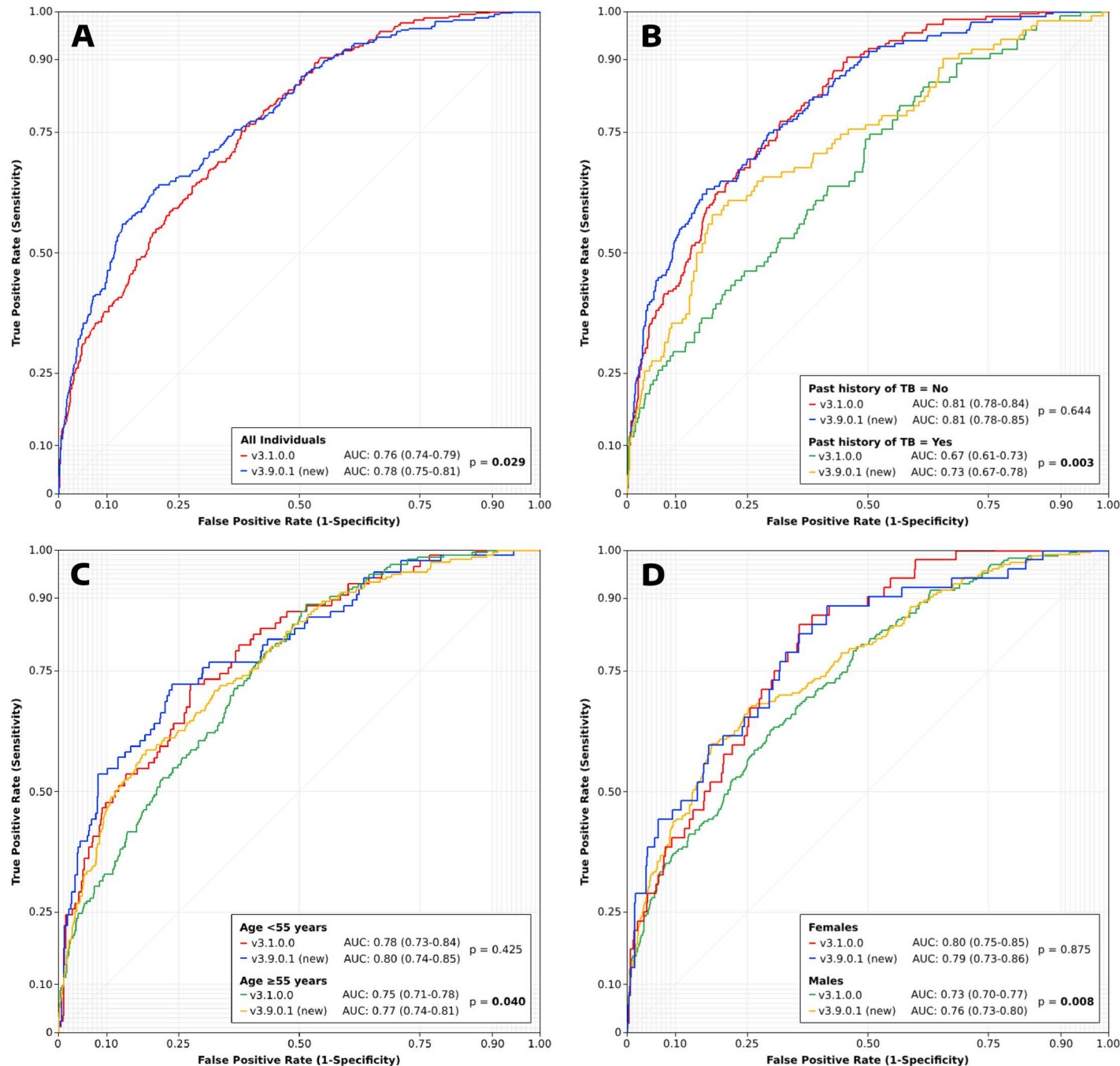

**Fig 1. ROC curves for the two Lunit CAD software versions for A) all individuals and stratified by B) past history of TB, C) age group and D) sex.**

Lunit CAD software versions, there was a significant increase in the median abnormality score which translates to significantly decreased specificity and thus significantly increased diagnostic testing indication if the abnormality score cut-off threshold is not re-calibrated for the new software version. This change in the distribution of abnormality scores has also been similarly reported in recent evaluations of other WHO-recommended CAD software platforms [20,21].

**Table 2. Comparison of the two Lunit CAD software in sub-cohorts.**

| | History of past TB | | | | No history of past TB | | | |
|---|---|---|---|---|---|---|---|---|
| | Age ≥55 years | | Age 15–54 years | | Age ≥55 years | | Age 15–54 years | |
| | Male | Female | Male | Female | Male | Female | Male | Female |
| Cohort size, n | 545 | 151 | 202 | 54 | 870 | 598 | 197 | 91 |
| MTB positive, n (%) | 62 (11.4%) | 11 (7.3%) | 26 (12.9%) | 3 (5.6%) | 93 (10.7%) | 29 (4.8%) | 47 (23.9%) | 9 (9.9%) |
| v3.1.0.0, AUC (95% CI) | 0.63 (0.56–0.70) | 0.71 (0.55–0.87) | 0.69 (0.56–0.82) | 0.90 (0.75–1.00) | 0.76 (0.72–0.81) | 0.80 (0.74–0.86) | 0.79 (0.73–0.86) | 0.84 (0.73–0.96) |
| v3.9.0.1 (new), AUC (95% CI) | 0.72 (0.65–0.79) | 0.74 (0.56–0.91) | 0.72 (0.60–0.85) | 0.81 (0.51–1.00) | 0.77 (0.71–0.82) | 0.80 (0.72–0.88) | 0.82 (0.75–0.88) | 0.83 (0.69–0.98) |
| p-value | **<0.001** | 0.725 | 0.260 | 0.299 | 0.895 | 0.959 | 0.209 | 0.848 |

**Table 3. Median (and interquartile range [IQR]) Lunit CAD abnormality scores by software version.**

| | All Individuals | p-value | Xpert MTB/RIF Ultra test result | | | |
|---|---|---|---|---|---|---|
| | | | MTB negative | p-value | MTB positive | p-value |
| v3.1.0.0 | 0.35 (0.05–0.73) | **<0.001** | 0.29 (0.04–0.69) | **<0.001** | 0.76 (0.48–0.91) | **<0.001** |
| v3.9.0.1 (new) | 0.61 (0.27–0.83) | | 0.57 (0.24–0.80) | | 0.89 (0.71–0.95) | |

**Table 4. Performance of the Lunit CAD software versions at different abnormality score cut-off thresholds.**

| | Cut-off Score | TP | FP | FN | TN | Sensitivity (95% CI) | Specificity (95% CI) | Change in Ultra testing | p-value |
|---|---|---|---|---|---|---|---|---|---|
| v3.1.0.0 | 0.217 | 253 | 1,315 | 27 | 1,113 | 90.4% (86.3–93.6) | 45.8% (43.8–47.9) | *Ref* | *Ref* |
| v3.9.0.1 (new) | 0.217 | 270 | 1,868 | 10 | 560 | 96.4% (93.5–98.3) | 23.1% (21.4–24.8) | +36.4% | **<0.001** |
| v3.9.0.1 (new) | 0.480 | 253 | 1,380 | 27 | 1,048 | 90.4% (86.3–93.6) | 43.2% (41.5–45.2) | +4.1% | 0.148 |

*TP = True Positive, FP = False Positive, FN = False Negative, TN = True Negative and 95% CI = 95% Confidence Interval*

It remains unclear how the new software version's AUC performance gains may translate into a real-world improvement in CXR interpretation accuracy during prospective TB screening. The evaluation's ROC curves indicate that the performance gains are localized in specific sub-cohorts, indicating there may be benefits to using the new software in these populations, which were not possible to assess with this test library due to its sample size. However, the ROC curves also indicate that these gains appear to be in abnormality score cut-off threshold ranges below what would normally be considered for threshold setting during TB screening. When comparing software sensitivity and specificity at a clinically-relevant abnormality score cut-off threshold (e.g., 90% sensitivity), we showed that the new software version's performance is actually not significantly different from its predecessor. However, in order to achieve this result, the new software's abnormality score cut-off threshold required re-calibration. There has been a lack of guidance from CAD software developers about how their software change with successive versions, and how TB programs should adjust their implementation as new software versions are rolled out [22]. Without such guidance, TB program implementers may continue to use their original threshold scores, potentially overwhelming laboratory capacity and reducing the cost effectiveness of their programs. This is particularly a risk when CAD software are used to extend CXR screening to settings without an on-site radiologist. Changes in the performance profile of a CAD software may undermine trust in and uptake of the technology.

The higher median abnormality scores in the newer software version and need for recalibration to maintain software performance are critical findings for CAD software implementers, who have likely gone through a process to determine the optimal abnormality score cut-off threshold for their original CAD software deployment. The WHO's Tropical Disease Research (TDR) unit recommends first conducting TB screening without CAD software, in order to generate a sufficiently powered data set to identify the optimal cut-off threshold for each setting [23]. Alternately, users may opt for an iterative threshold score calibration (ITSC) method which allows for continuous cut-off threshold adjustment during a programmatic CAD software deployment until the software's target performance is achieved [24]. Both of these methods for cut-off threshold calibration represent a burden to TB programs, which evidence suggests may now need to be replicated as new CAD software versions are deployed. CAD software developers must be more transparent about the underlying changes to their software which have been made between software version updates. Certain software updates focused on turnaround-time or user interfaces are unlikely to have an impact on CXR interpretation accuracy, and could be rapidly adopted. However, changes to the underlying algorithm that affect CXR interpretation accuracy to date have only been discovered through investigation by the TB program implementers. Programs are unlikely to run two CAD software versions in parallel, so investigations are limited to pre/post analysis of data or retrospective test library evaluations, such as this one.

This CAD software evaluation is not without limitations. One key decision when creating a DICOM test library is the selection of a reference standard. In this evaluation's test library, all individuals have a valid Ultra test result as the microbiological reference standard. Since the Ultra assay is not 100% sensitive, it is possible that some individuals in the test library may have a false negative result. In addition, during programmatic ACF, the vast majority of people screened by CXR will never be indicated for follow-on diagnostic testing. Thus, this test library under samples individuals who have a normal CXR result and who would also likely have a low abnormality score on a CAD software. As a result, this test library cannot be used to assess a CAD software's performance against the Target Product Profile criteria for a TB triage test [25] or to identify an optimal cut-off threshold for programmatic implementation. Despite these limitations, this test library design is common among published CAD software performance evaluations [12–15,17], Future studies should consider building DICOM test libraries which use a composite reference standard with a mix of radiological and diagnostic test results, in order to more accurately reflect the demographics of the participants being screened during programmatic ACF.

In this test library, individuals with a past history of TB were tested using the Ultra assay. The literature shows that it is possible to have a false positive Ultra test result many years after completion of TB treatment [26]. We do not have data on the time since a previous episode of TB, so it is possible that some individuals included in this test library may have a false positive Ultra test result. This may result in an overstatement of the CAD software's performance in this sub-cohort. However, other CAD software evaluations have included similar individuals in their test libraries [12,14,17,27] and in this test library, there is no significant difference in the Ultra test positivity rates between individuals with and without a history of TB.

## Materials and methods

### Ethics statement

The programmatic ACF initiatives which generated the data from which the DICOM test library was constituted were implemented within Ministry of Health of Viet Nam guidelines for TB screening (1314/QĐ-BYT) and received implementation approval from the Ho Chi

Minh City People's Committee (Decision No. 2681/QĐ-UBND). Since this programmatic ACF was within national TB screening guidelines, only verbal consent was obtained from participants. Ethical approval for this de-identified retrospective CAD software evaluation was granted by Pham Ngoc Thach Hospital (Decision No. 1688/HĐĐĐ-PNT).

## TB screening activities

Friends for International TB Relief (FIT) implemented 127 mobile CXR screening events during programmatic active case finding (ACF) across Ho Chi Minh City, Viet Nam (Districts 6, 8, 12, Binh Chanh, Go Vap, Hoc Mon and Tan Binh) between March 2019 and March 2021. The methods for participant mobilization and evaluation have been previously described in the literature [28,29]. Briefly, individuals with an increased risk of having TB (e.g., contacts, older people [≥55 years old], those with TB symptoms, those with limited access to health facilities, etc.) were mobilized in advance of the screening events. All individuals were evaluated using a verbal symptom questionnaire loaded onto a custom-built mHealth app and CXR in parallel. Three different radiography systems were used to capture the CXR images: DRTECH, South Korea; Konika Minolta, Japan; and JPI Healthcare, United States. An on-site radiologist then read and classified each CXR image as 'Normal', 'Abnormal – TB' or 'Abnormal – Other'. Individuals with an 'Abnormal – TB' CXR result were asked to provide a sputum specimen for testing with the Ultra assay; at a subset of screening events, individuals with an 'Abnormal – Other' CXR result may have also been indicated for diagnostic testing at the radiologist's discretion. At the end of each screening event, DICOM files were extracted from the radiography system and archived on a hard drive stored at the FIT office.

## DICOM test library creation

FIT's ACF initiative screened 30,135 people using CXR, resulting in a total of 3,231 being tested by the Ultra assay (Fig 2). 305 (9.4%) individuals had a MTB positive test result, 2,855 (88.4%) had a MTB negative test result and 71 (2.2%) had a failed test result (e.g., Error, Invalid or No Result outcome) without being retested. 281 (92.1%) individuals with a MTB positive test result and 2,429 (85.1%) with a MTB negative test result were then selected for inclusion in the final DICOM test library, based on the completeness of the demographic data exported from the program's mHealth system, their age (≥18 years) and the availability of their DICOM files.

## CAD software processing

The DICOM files included in the test library were fully de-identified using a python tool built by FIT. The python tool removed DICOM attributes (e.g., participant name, year of birth, gender, acquisition date and time, etc.) and tags (e.g., X-ray manufacturer, site name, etc.) were stripped from the file and the file was re-labeled with a study identifier. The de-identified DICOM files were then securely transferred to Lunit for blinded processing with their INSIGHT CXR CAD software v3.1.0.0 and the newer v3.9.0.1. Paired continuous CAD software abnormality scores were returned to FIT for analysis.

## Statistical analyses

Descriptive statistics for the DICOM test library were prepared and compared by Ultra test result using a chi-squared test. The AUCs were calculated for each software version and compared using the *roccomp* command [30] in Stata version 17 (College Station, Texas, United States). The AUCs were compared between software versions by participant demographics to identify localized software improvements in key sub-populations, including people with a

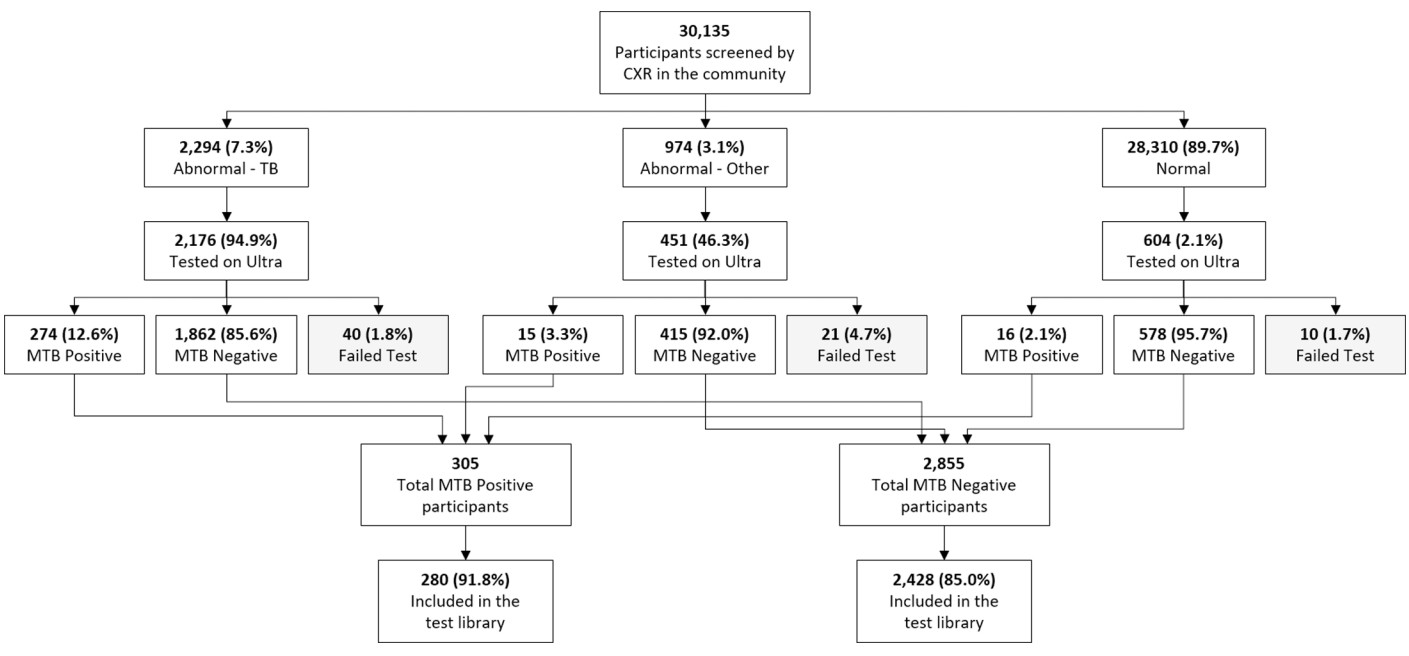

**Fig 2. Flow diagram of TB screening and constitution of the DICOM test library.**

history of TB, males and older individuals. The median Lunit CAD abnormality scores were calculated compared across various clinical and demographic sub-groups using a Wilcoxon sign rank test, and the proportion of people with a MTB positive test result and an abnormality score <0.50 (default cut-off threshold) was calculated for each software version. Finally, Lunit CAD software performance was calculated and compared using the *roccomp* command at a threshold score corresponding to 90% sensitivity for the earlier software version. The software versions were then compared at thresholds which corresponded to the sensitivity of the new software when the original threshold score was used.

## Acknowledgments

We would like to thank the Viet Nam National Tuberculosis Programme, Pham Ngoc Thach Hospital and the District TB Units in Districts 6, 8, 12, Binh Chanh, Go Vap, Hoc Mon and Tan Binh for their support with the implementation of community-based CXR screening activities. We would also like to thank Lunit for processing the test library's DICOM files with their INSIGHT CXR CAD software free of charge.

## Author contributions

**Conceptualization:** Andrew James Codlin, Luan Nguyen Quang Vo, Rachel Forse, Hoa Binh Nguyen.

**Data curation:** Andrew James Codlin, Thang Dao.

**Formal analysis:** Andrew James Codlin.

**Funding acquisition:** Andrew James Codlin, Luan Nguyen Quang Vo, Rachel Forse.

**Investigation:** Andrew James Codlin, Luan Nguyen Quang Vo, Thang Dao, Rachel Forse.

**Methodology:** Andrew James Codlin, Kristi Sidney Annerstedt, Johan Lundin, Knut Lönnroth.

**Project administration:** Andrew James Codlin, Luan Nguyen Quang Vo, Thang Dao, Rachel Forse, Ha Thi Minh Dang, Lan Huu Nguyen, Hoa Binh Nguyen, Luong Van Dinh.

**Resources:** Andrew James Codlin, Luan Nguyen Quang Vo, Rachel Forse, Ha Thi Minh Dang, Lan Huu Nguyen, Hoa Binh Nguyen, Luong Van Dinh.

**Supervision:** Andrew James Codlin, Luan Nguyen Quang Vo, Rachel Forse, Kristi Sidney Annerstedt, Johan Lundin, Knut Lönnroth.

**Validation:** Andrew James Codlin, Thang Dao, Rachel Forse.

**Visualization:** Andrew James Codlin, Kristi Sidney Annerstedt, Johan Lundin, Knut Lönnroth.

**Writing – original draft:** Andrew James Codlin.

**Writing – review & editing:** Andrew James Codlin, Luan Nguyen Quang Vo, Thang Dao, Rachel Forse, Ha Thi Minh Dang, Lan Huu Nguyen, Hoa Binh Nguyen, Luong Van Dinh, Kristi Sidney Annerstedt, Johan Lundin, Knut Lönnroth.

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
