## [Decision Letter · Decision Letter 0]

31 Oct 2024

PDIG-D-24-00271

Comparison of different Lunit INSIGHT CXR software versions when reading chest radiographs for tuberculosis

PLOS Digital Health Dear Dr. Codlin, Thank you for submitting your manuscript to PLOS Digital Health. After careful consideration, we feel that it has merit but does not fully meet PLOS Digital Health's publication criteria as it currently stands. Therefore, we invite you to submit a revised version of the manuscript that addresses the points raised during the review process. Please submit your revised manuscript within 60 days. If you will need more time than this to complete your revisions, please reply to this message or contact the journal office at digitalhealth@plos.org. Please include the following items when submitting your revised manuscript: * A rebuttal letter that responds to each point raised by the editor and reviewer(s). You should upload this letter as a separate file labeled 'Response to Reviewers '. This file does not need to include responses to any formatting updates and technical items listed in the 'Journal Requirements' section below.* A marked-up copy of your manuscript that highlights changes made to the original version. You should upload this as a separate file labeled 'Revised Manuscript with Track Changes '.* An unmarked version of your revised paper without tracked changes. You should upload this as a separate file labeled 'Manuscript '. If you would like to make changes to your financial disclosure, competing interests statement, or data availability statement, please make these updates within the submission form at the time of resubmission. Guidelines for resubmitting your figure files are available below the reviewer comments at the end of this letter. We look forward to receiving your revised manuscript. Kind regards, Md. Mehedi HassanAcademic EditorPLOS Digital Health Leo Anthony CeliEditor-in-ChiefPLOS Digital Healthorcid.org/0000-0001-6712-6626 **Journal Requirements:**

1. In the online submission form, you indicated that "The test library’s anonymized DICOM files may be shared upon a reasonable request to the corresponding author.". 

3. Uploaded as supplementary information.

2. In the online submission form, you indicated that your data will be submitted to the Dryad database upon acceptance. Should your submission be accepted, we will require the following information in your Data Availability Statement: 

 1. The DOI provided by Dryad

 2. The citation for your data package in the reference section of your manuscript

 3. The citation for your data package in the methods section

 If you are unable to adhere to our open data policy, please kindly revise your statement to explain your reasoning and we will seek the editor's input on an exemption.

**Reviewers' Comments:**Reviewer's Responses to Questions

**Comments to the Author**

1. Does this manuscript meet PLOS Digital Health’s publication criteria ? Is the manuscript technically sound, and do the data support the conclusions? The manuscript must describe methodologically and ethically rigorous research with conclusions that are appropriately drawn based on the data presented.

Reviewer #1: Yes

Reviewer #2: Yes

Reviewer #3: Partly

2. Has the statistical analysis been performed appropriately and rigorously?

Reviewer #1: Yes

Reviewer #2: Yes

Reviewer #3: No

3. Have the authors made all data underlying the findings in their manuscript fully available (please refer to the Data Availability Statement at the start of the manuscript PDF file)?

Reviewer #1: Yes

Reviewer #2: Yes

Reviewer #3: No

4. Is the manuscript presented in an intelligible fashion and written in standard English?

Reviewer #1: Yes

Reviewer #2: Yes

Reviewer #3: Yes

5. Review Comments to the Author

Reviewer #1: Introduction: The introduction effectively sets the stage for the study, providing context about the importance of CXR in TB screening and the role of CAD software. It might be beneficial to expand slightly on the limitations of previous CAD software studies to highlight the novelty of the current research.

Methods: The methods section is detailed and clear. The use of a well-characterized DICOM test library and the consideration of various clinical and demographic variables add strength to the study. It would be helpful to include more details on the ethical considerations and consent processes, particularly given the retrospective nature of the study.

Results: The results are well-presented with appropriate use of tables and figures. The significant findings are clearly highlighted. The discussion of AUCs and abnormality scores is thorough, though further elaboration on the potential clinical implications of these findings could be beneficial.

Discussion: The discussion is insightful, addressing the strengths and limitations of the study. The authors correctly identify the need for prospective studies to assess real-world improvements. Expanding on how these findings might influence TB screening protocols and CAD software deployment could provide additional value.

Items to consider:

1. Ensure that all abbreviations are defined at first use.

Just One Example for Reference :

Original Text:

"We independently measured the differences in CAD software performance between INSIGHT CXR (Lunit, South Korea) versions 3.1.0.0 and 3.9.0.1."

Corrected Text:

"We independently measured the differences in computer-aided detection (CAD) software performance between INSIGHT chest X-ray (CXR) (Lunit, South Korea) versions 3.1.0.0 and 3.9.0.1."

2. Correct any typographical or grammatical errors.

Just One Example for Reference

Original Text:

"Chest X-rays (CXRs) are a highly sensitive tool for tuberculosis (TB) screening."

Corrected Text:

"Chest X-rays (CXRs) are highly sensitive tools for tuberculosis (TB) screening."

3. Ensure that references are correctly formatted and relevant to the study.

Original Reference:

Just One Example for Reference

"1. Law I, Floyd K, Group the ATPS. National tuberculosis prevalence surveys in Africa, 2008–2016: an overview of results and lessons learned. Tropical Medicine & International Health. 2020;25: 1308–1327. doi:10.1111/tmi.13485"

Corrected Reference:

"1. Law I, Floyd K, the African TB Prevalence Survey Group. National tuberculosis prevalence surveys in Africa, 2008–2016: an overview of results and lessons learned. Tropical Medicine & International Health. 2020;25(12):1308-1327. doi:10.1111/tmi.13485."

Reviewer #2: Codlin et al have written an excelent manuscript about the performance of different versions of Lunit’s CAD software which is important for understanding how this type of software chnages and is needed especially for Lunit since it was part of the original WHO recommendations for TB but has been part of relatively few peer reviewed publications compared to qXR and CAD4TB. The authors should be congratulated for a well written and clear piece.

I have several minor comments and suggestions which should be addressed before the manuscript is accepted and one major revision although I hope it will not be very difficult to update.

The summary abstract in the generated PDF (I assume where you enter the data in the online system is slightly different than the manuscript abstract and results – one mentions 10.4% and then in the manuscript 10.3% is the overall positivity rate for the sample.

Lines 12-15 sometimes the new (better performing) AUC number is listed first and sometimes it is second, making it confusing for the reader. I would just list the newer and higher number first so it flows better.

Line 50. We often talk about design locked products for diagnostics and I think what the authors are referring to is that the CAD products change more regularly than other TB diagnostics for example Xpert. However, Xpert MTB/RIF changed during the time it was produced – and each of those chnages was a new design locked product. In the same vein, 3.1.0.0 and 3.9.0.1 are locked. They do not change based on their data they interpret, and they are CE marked for that specific version. It gives the wrong impression to a reader in the current wording and should be changed to something like, as opposed to laboratory-based diagnostics, CAD software is updated regularly with new versions going through separate regulatory approval... or something to that effect.

Lines 52-53 – I generally agree with the statement but your reference 8 from Murphy et al actually showed that V4 was marginally better than V5 for AUC – although V6 outperformed them both.

Line 63 – positive should be with a lowercase p.

65-66 you might consider mentioning how many older/younger individuals tested positive for TB in addition to the yield.

82 I very much like the point about the performance gain in the lower levels of sensitivity which is quite important and not often discussed in CAD publications.

145-146. However, when an abnormality score cut-off threshold is set in order to maximize sensitivity, the new software version appears to be significantly less specific than its predecessor – i am not sure this manuscript is showing this to be true – as a comparison of the two software which leads me to my major suggestion for the submission.

The analysis compares the performance of V3.9 at the same threshold that 3.1 achieved 90% sensitivity. This is a good comparison as it demonstrates the importance of understanding how versions differ. Some changes are small – V3 and 4 of qXR and some are very large as Fehr and colleagues pointed out with CAD4TB 5,6, and 7 and they have real consequences for implementers who do not pay attention to the changes. However, as a full evaluation the authors should make the comparison at the sensitivity level – not only same threshold. Although it is implied in the AUC mentioned above in line 82, and then again in 140 but it is not explicit. The authors do present improved in sub-populations in Table 2 which is also nice to show improvement but the same could be done with how AUC translates to people detected/tests conducted/saved. I think a simple table could show the reader if a program wanted to fix sensitivity at 90% for either version, what would it mean for performance gain/loss tests used etc. This is critical for implementers to keep in mind. Some providers are offering perpetual licenses. Often, to upgrade there may be cost implications. So does it make sense for a program to spend money to upgrade if the performance gain would be minimal? Especially if deploying on many sites. I understand the library is curated and not reflective – but the analysis and approach is the same and I suggest be presented and discussed o make the statement on lines 141-75 stronger and easier to interpret.

148-150. I am not sure why humans are mentioned here since they are outside of the analysis and is pure supposition – both for the newer and older versions.

160-163. While I understand the point of the statement, I don’t think it is a strong one. Presumably, if as stated in lines 153-54 an implementer (ideally) would have gone through an analysis and evaluation of the software to arrive at a threshold score to select. The same implementer could use the same images to analyze with the new software. The point is that they must do it to ensure optimal performance, but I don’t think the statements in 153 and 160 can both be true. The authors shoud reinforce the importance of doing such analysis at before large scale implementation as well as with any software update and that developers should be more up front about the chnages they make and how it could affect performance at different thresholds.

Reviewer #3: This manuscript addresses an important aspect of TB diagnostics in the context of evolving AI technologies. The study design is compact, and the findings are relevant to both research and clinical practice. However, several improvements in the clarity of the abstract, deeper methodological discussions, and expanded clinical implications would enhance the manuscript’s overall impact.

1. The abstract needs to be rewritten. The objective of the study is not clearly stated and jumps directly into the contributions and results.

2. The phrase "performance characteristics were compared at selected cut-off thresholds" is unclear—briefly explain why these thresholds were chosen.

3. The introduction needs more details on why comparing CAD versions is important for clinical practice, especially in resource-constrained settings. While the shortage of radiologists is mentioned, explain how the findings could directly impact health outcomes.

4. Add more recent studies to highlight developments in CAD systems beyond the three mentioned systems (CAD4TB, INSIGHT CXR, qXR).

5. Add more recent studies on the clinical impact of CAD software in TB diagnostics within the field.

6. The limitations section should explain in more detail how using the Ultra test as a reference could bias the results. Although the issue of false positives in individuals with a past TB history is mentioned, expanding on the potential for this bias and its effect on the study’s conclusions is necessary.

7. Highlight the need for prospective validation more clearly. The differences between study conditions and real-world settings should be expanded.

8. In the methods section, how does the current threshold calibration method (TDR or iterative score calibration) compare in terms of ease of implementation, costs, and time?

9. Provide further details on the anonymization of DICOM files for external processing to enhance transparency regarding data security and patient confidentiality.

10. The figures, particularly the ROC curves (Figure 1), are critical for understanding the software's performance. Consider adding interpretative comments directly on the figures or in the captions.

11. The comparison between sub-cohorts in Table 2 is a strong feature of this manuscript. However, is this a result of insufficient power for smaller groups? Would a larger study be able to identify more nuanced differences?

12. The discussion focuses heavily on the statistical results without fully addressing the clinical significance of the findings. For instance, while the newer software's AUC is statistically higher, what does this mean for TB screening programs in practical terms? Could these gains in AUC translate into earlier diagnosis or better case management?

13. The sensitivity and specificity trade-offs are important but need further elaboration. Discuss how recalibrating thresholds will impact operations.

14. Consider discussing the implications of this study for low- and middle-income countries in more detail. Could the increased diagnostic testing triggered by a higher sensitivity software version strain health resources, and how should programs prepare for this?

15. Standardize the abbreviation for "AUC" throughout the manuscript (sometimes referred to as "ROC AUC").

16. Ensure consistent use of hyphens and spacing, e.g., in “past-history” and “past history” of TB.

17. The median abnormality score data could be better visualized by adding a table for easy comparison across different subgroups.

6. PLOS authors have the option to publish the peer review history of their article (what does this mean? ). If published, this will include your full peer review and any attached files.

**Do you want your identity to be public for this peer review?** For information about this choice, including consent withdrawal, please see our Privacy Policy .

Reviewer #1: **Yes**

Reviewer #2: No

Reviewer #3: **Yes**

---

## [Decision Letter · Decision Letter 1]

28 Jan 2025

PDIG-D-24-00271R1Comparison of different Lunit INSIGHT CXR software versions when reading chest radiographs for tuberculosisPLOS Digital Health Dear Dr. Codlin, Thank you for submitting your manuscript to PLOS Digital Health. After careful consideration, we feel that it has merit but does not fully meet PLOS Digital Health's publication criteria as it currently stands. Therefore, we invite you to submit a revised version of the manuscript that addresses the points raised during the review process. Please submit your revised manuscript within 30 days Feb 27 2025 11:59PM. If you will need more time than this to complete your revisions, please reply to this message or contact the journal office at digitalhealth@plos.org. Please include the following items when submitting your revised manuscript:* A rebuttal letter that responds to each point raised by the editor and reviewer(s). You should upload this letter as a separate file labeled 'Response to Reviewers '. This file does not need to include responses to any formatting updates and technical items listed in the 'Journal Requirements' section below.* A marked-up copy of your manuscript that highlights changes made to the original version. You should upload this as a separate file labeled 'Revised Manuscript with Track Changes '.* An unmarked version of your revised paper without tracked changes. You should upload this as a separate file labeled 'Manuscript '. If you would like to make changes to your financial disclosure, competing interests statement, or data availability statement, please make these updates within the submission form at the time of resubmission. Guidelines for resubmitting your figure files are available below the reviewer comments at the end of this letter. We look forward to receiving your revised manuscript. Kind regards, Peter H Charlton, MEng, PhDSection EditorPLOS Digital Health Peter H CharltonSection EditorPLOS Digital Health Leo Anthony CeliEditor-in-ChiefPLOS Digital Healthorcid.org/0000-0001-6712-6626 **Additional Editor Comments (if provided):** Thank you for revising this manuscript. Please see the remaining minor comments from Reviewer 3 about adding information on the clinical impact of CAD systems, and improving clarity. I believe addressing these would improve the manuscript.**Reviewers' Comments:** Reviewer's Responses to Questions

**Comments to the Author**

1. If the authors have adequately addressed your comments raised in a previous round of review and you feel that this manuscript is now acceptable for publication, you may indicate that here to bypass the “Comments to the Author” section, enter your conflict of interest statement in the “Confidential to Editor” section, and submit your "Accept" recommendation.

Reviewer #2: All comments have been addressed

Reviewer #3: All comments have been addressed

2. Does this manuscript meet PLOS Digital Health’s publication criteria ? Is the manuscript technically sound, and do the data support the conclusions? The manuscript must describe methodologically and ethically rigorous research with conclusions that are appropriately drawn based on the data presented.

Reviewer #2: Yes

Reviewer #3: Yes

3. Has the statistical analysis been performed appropriately and rigorously?

Reviewer #2: Yes

Reviewer #3: Yes

4. Have the authors made all data underlying the findings in their manuscript fully available (please refer to the Data Availability Statement at the start of the manuscript PDF file)?

Reviewer #2: No

Reviewer #3: Yes

5. Is the manuscript presented in an intelligible fashion and written in standard English?

Reviewer #2: Yes

Reviewer #3: Yes

6. Review Comments to the Author

Reviewer #2: (No Response)

Reviewer #3: The authors have made significant revisions that address most of the points raised. However, some areas, such as the inclusion of recent studies on the clinical impact of CAD systems, could be further expanded. Additionally, adding interpretative comments on figures and visualizing median scores in a table would improve clarity.

Despite these minor suggestions, the manuscript has improved considerably, and I believe it is now suitable for publication with minor revisions.

7. PLOS authors have the option to publish the peer review history of their article (what does this mean? ). If published, this will include your full peer review and any attached files.

**Do you want your identity to be public for this peer review?** For information about this choice, including consent withdrawal, please see our Privacy Policy .

Reviewer #2: No

Reviewer #3: **Yes: ** Farhana Yasmin

---

## [Editor Report · Decision Letter 2]

5 Mar 2025

Comparison of different Lunit INSIGHT CXR software versions when reading chest radiographs for tuberculosis

PDIG-D-24-00271R2

Dear Mr Codlin,

We are pleased to inform you that your manuscript 'Comparison of different Lunit INSIGHT CXR software versions when reading chest radiographs for tuberculosis' has been provisionally accepted for publication in PLOS Digital Health.

Best regards,

Peter H Charlton, MEng, PhD

Section Editor

PLOS Digital Health

**Additional Editor Comments (if provided):**

Thank you for addressing the comments